# Factors Associated with the Adoption of Drones for Product Delivery in the Context of the COVID-19 Pandemic in Medellín, Colombia

Alejandro Valencia-Arias [1,*], Paula Andrea Rodríguez-Correa [2], Juan Camilo Patiño-Vanegas [3], Martha Benjumea-Arias [1], Jhony De La Cruz-Vargas [4] and Gustavo Moreno-López [5]

1   Facultad de Ciencias Económicas y Administrativas, Instituto Tecnológico Metropolitano, Medellín 050034, Colombia
2   Centro de Investigaciones, Institución Universitaria Escolme, Medellín 050012, Colombia
3   Facultad de Negocios Internacionales, Universidad Santo Tomás, Medellín 050041, Colombia
4   Director general, Instituto de Investigación en Ciencias Biomedicas, Universidad Ricardo Palma, Lima 15039, Peru
5   Grupo de Investigación en Educación y Ciencias Sociales y Humanas, Institución Universitaria Marco Fidel Suárez, Bello 051050, Colombia
*   Correspondence: jhoanyvalencia@itm.edu.co; Tel.: +57-(300)-256-7977

**Abstract:** This study aims to identify the factors associated with the adoption of drone delivery in Medellín, Colombia, in the context of the COVID-19 pandemic. For that purpose, it implemented the Diffusion of Innovation (DOI) theory and the Technology Acceptance Model (TAM), which have constructs that complement each other to determine the decision to accept a given technology. A survey was administered to 121 participants in order to validate the model proposed here, which is based on variables that reflect the perceived attributes and risks of this innovation and individuals' characteristics. The results indicate that the factors Performance Risk, Compatibility, Personal Innovativeness, and Relative Advantage of Environmental Friendliness have the greatest influence on Intention to Use Drone Delivery (mediated by Attitude Towards Drone Delivery). This paper offers relevant information for the academic community and delivery companies because few other studies have investigated this topic. Additionally, the proposed technology adoption model can be a benchmark for other emerging economies in similar social, economic, and technological conditions.

**Keywords:** drones; drone delivery services; contactless delivery strategies; COVID-19; pandemic

## 1. Introduction

In a globalized and competitive world, innovation becomes a strategy for companies to remain in the market [1]. However, some factors limit their success; for example, end consumers are still reluctant to migrate to some new technologies or trends and question whether it is advisable or not to adopt them because they are satisfied with the services they have traditionally received and feel that adopting these new technologies is irrelevant [2]. In the present day, companies adopt technologies to improve production, sales, and logistics processes due to their multiple advantages. The number of logistics operations of delivery services has been increased by the growing volume of online orders. As a result of this expansion and the different alternatives in the market, customers raise their expectations regarding high-quality, faster delivery, for which they are willing to pay a premium price [3].

Consequently, companies are constantly investing in the search for innovative solutions to improve their delivery systems that enhance their effectiveness and are environmentally friendly [4]. For example, they have explored electric vehicles, artificial vision, and machine learning for autonomous vehicles [5]. Taking into account the current context of COVID-19 pandemic, the use of drones for parcel delivery is expected to increase in the foreseeable future [6]. This is especially true considering that the current crisis is generated

by the COVID-19 virus, which spreads mainly through the respiratory system of infected individuals and has led to social distancing as a strategy to reduce the risks of spreading it [7]. Thus, autonomous vehicles such as drones present a great opportunity for food or even drug delivery, offering a possible solution to the current problems caused by the presence of COVID-19 [8]. For instance, drones have been shuttling medicine and samples from suspected COVID-19 patients for testing [9] to difficult-to-access areas in developing countries, such as Ghana [10].

During the 2020–2021 COVID-19 lockdowns in Medellín, Colombia, a startup company called Rappi developed a special way of delivering its orders. It deployed a fleet of robots, built by Kiwibot, to deliver takeout food to people in lockdown [11]. In the same city, drones have been used to generate landslide risk mitigation strategies in low-income settlements [12] and as support for the thermal analysis of urban environments, facilitating the analysis of urban heat islands [13]. This reflects the recent acceptance and adaptation of drones in different services in Medellín. Therefore, we should analyze the issue of merchandise delivery employing these technologies in said city.

Due to the crisis generated by the COVID-19 pandemic, Medellín faced challenges in different sectors, but especially in health care services. For example, during the contingency, medicine and pharmaceutical care were provided in person. However, at the same time, mobility restrictions were imposed, the demand for medicine increased, and there was a countrywide shortage of medical products. Thus, the national government established vulnerability criteria, prioritization strategies, and restrictions for the general population. In such situations, it was necessary to establish new home care channels or models to strengthen self-care supported by technological tools [14].

In a more global context, it has become necessary to generate mechanisms to prevent contagion in everyday activities, such as studying, working, or buying from home, which are possible thanks to recent technological developments [15]. Regarding online purchasing, recent data from 2022 [16] indicate a significant growth in the volume of scientific literature on the logistics of e-commerce. Approximately 56% of the articles about this topic have been published in the last three years.

Due to the growth of e-commerce, the logistics market is being confronted by challenges and requirements brought about by digitization [17]. For example, according to Statista (2019) as cited in [18], worldwide e-commerce sales reached 3.53 trillion US dollars in 2019. To deliver that volume of orders, drivers and service providers strive to provide adequate customer service [19]. However, difficulties arise on a daily basis (e.g., delayed or broken packages, stressed employees, and angry customers).

Consumers have become more demanding regarding these inconveniences, and, due to their faster pace of life, they require delivery that is timely (among other characteristics) [20]. Therefore, companies that use e-commerce seek to meet their expectations by ensuring responsiveness, while optimizing their resources in terms of costs and time [21]. Technological advances present both good and bad aspects, but as mankind adapts itself and interacts with them and the technology is improved, they turn out to be beneficial [22].

Different technologies can be implemented to respond to these challenges in delivery services, and drones are one of them. A drone is an aircraft that can be remotely flown without a human pilot [21]. The use of drones has advantages and disadvantages perceived by both user companies and end consumers [23]. However, this type of technology has a high potential for the commercial sector due to its qualities in terms of speed, cost, safety, and minimal human intervention [24]. In the context of the COVID-19 pandemic, Euchi [8] identified advantages in drones, such as disinfection (which reduces the risk of contagion) and social distancing (because they are remotely controlled). They can also transport samples of suspected COVID-19 patients, once again contributing to social distancing [25].

Jiang and Ren (2020) [20] proposed a prospect theory that takes into account the factors that support the superiority of drones over manned aircrafts. Such factors include delivery distance, degree of rider delay, pickup time, and consumer attitudes towards drone delivery. However, it is clear to them that this is a very vast field yet to be explored [26]. For example,

the weather can pose a threat to their normal operation, and the layout of drone airports should be further analyzed. Raj and Sah (2019) [21] consider it important to investigate the critical success factors for this kind of technology in the logistics sector, its technical aspects, availability of skilled workforce, and government policies.

During the pandemic due to the outbreak of the COVID-19 virus, drones were proposed as an innovative tool with a vast potential to reduce the risk of contagion in that exceptional context marked by social distancing. Thus, to bridge the gaps produced by physical separation, drones have been employed to respond to specific challenges related to the pandemic (e.g., disinfection, delivery, and surveillance) [27] in several countries, but not necessarily everywhere or in the same way.

For example, in India a mechanism was proposed to effectively improve the process of treating COVID-19 patients by implementing drone services to reduce the risk of infection of doctors or other medical staff, thus preventing the spread of the infection [28]. In Spain, a study [29] evaluated the possibility of using drones for disinfection tasks in outdoor public service areas to reduce virus transmission. In Ireland, these systems have been used to combat COVID-19 through monitoring and detection, social distancing, disinfection, data analysis, and delivery of goods and medical supplies [30]. In Turkey, since the virus can be easily transmitted from person to person, retailers have started testing drones to deliver products ordered online. Therefore, drones are indeed an alternative delivery system that could solve some of these problems in different regions [31].

Although drone delivery during the pandemic has been researched in some developed countries [7,31,32], few studies have addressed this phenomenon in Latin America. Therefore, as stated above, this study aims to (1) analyze the factors associated with the adoption of drones for goods delivery in the context of the COVID-19 pandemic in Medellín and (2) present an overview of how this service is perceived in a developing economy.

## 2. Narrative Literature Review

Using drones as vehicles for cargo delivery is an opportunity for economic and environmental development and establishes a new sustainable business model [33]. Drone delivery is based on machine learning and artificial intelligence technologies that require a high initial investment in terms of skilled workforce, technicians, and fulfillment centers [21], as well as the construction of infrastructure known as drone airports [34].

One of the main advantages perceived by end consumers in drone delivery is environmental protection through environment-friendly products and the reduction in air and noise pollution, for which they are willing to pay a higher price [21]. The environmental friendliness of this technology is a key factor influencing and motivating its adoption [35]. Therefore, it is very important to raise consumer awareness in this regard based on research on their behavior in terms of values, beliefs, and social norms [36]. The environmental education of consumers is a step toward ensuring the preservation of the environment [37].

In addition to ecological advantages, drones offer benefits such as cheaper, faster shipping [38], safety, speed, environmental friendliness, and convenience. Thus, they could replace traditional transportation in parcel delivery services [39]. Drones are not affected by traffic jams or heavy traffic on roads. They are operated by a computer system that can reduce labor costs, which is a clear advantage for sustainability [37]. However, in the post-COVID-19 period, consumers may radically change their behavior in terms of their attitude towards and intention to use drones for the delivery of goods such as food or medicine [7].

The advantages of drones over traditional means in logistics are evident, especially their significant reduction in package delivery time and increased reliability, efficiency, security, and stability. However, as explained by Sah et al. (2021) [40], the widespread implementation of this disruptive logistics technology is not yet visible. The most relevant barriers for the implementation of drones in logistics are related to a greater extent to the regulations of each country and the threats they might pose to individual privacy and security. Other barriers include public perception and environmental, technical, and

economic aspects. Additionally, not all types of suppliers can use drones to provide their logistics services. Thus, it may be impractical to implement a finely differentiated delivery strategy [41] because, for that purpose, logistics providers should cooperate more intensely, and the flow of goods needs to be further consolidated [42].

Despite the potential advantages of drones, users also perceive their risks and disadvantages. For example, as they are remotely controlled by computers, they are exposed to cyberattacks. Also, people fear for their privacy due to the possibility of being recorded or attacked by drones [43]. Therefore, detailed user knowledge of drones and their functions is a key factor for their adoption [21]. Consumer reactions to drone delivery indicate that they resist the change and the adoption of new technologies because of their strong belief that the traditional system is safer. However, the credibility of certain brands and consumers' trust in them influences the adoption and acceptance of drones [3].

To use drones for food delivery services, we should consider different risks associated with it. As explained by Mathew et al. (2021) [44], consumers may perceive risk in a new technology-mediated product/service due to ambiguity or lack of credibility. Three main types of risks are evident in food delivery: performance, delivery, and privacy. Thus, the image of drone food delivery services tends to be affected by perceived risks stemming from concerns about the use of new technologies. Said concerns also refer to financial and psychological risks. Performance risk reflects consumers' concerns about losses incurred when the service does not work as expected, especially in times of COVID-19; thus, they cannot make accurate performance decisions before using the service [45,46].

Other perceived disadvantages regarding the functionality of drones are their batteries and flight duration, due to which logistics centers or airports would have to be built at certain distances. Their maximum weight capacity is also a limiting factor for the provision of the delivery service because it is usually 5 kg [3]. In addition, some other external factors may cause accidents, such as falling from heights; colliding with trees, buildings, animals, power lines [38], or drones from other companies; and weather conditions that prevent the provision of the service [5–7]. As a result, designating special airways for drones is essential. Nevertheless, some countries lack national policies to regulate drone logistics for delivery services [47].

Companies have found physical and financial risks in drones that result in drawbacks for their adoption. In addition, end consumers have expressed that social interaction (which they would not have with drones) is important in the provision of the service [17]. Such beliefs regarding the risks of product delivery methods vary among consumers. According to Zhu (2019) [48], exploring consumer behavior and profiles and conducting communication campaigns contribute immensely to the acceptance of commercial drone delivery. In general, the estimates of user acceptance range between great skepticism and exaggerated optimism.

Although some companies are already using autonomous vehicles in pilot tests of delivery services for e-commerce [5], safety and privacy are still a concern for end consumers. Therefore, it is necessary to inquire about their intention to adopt or oppose the use of this kind of service, especially in the context of a pandemic in which precisely social distancing was encouraged to reduce the risks of spreading COVID-19 [7]. The business world is actively considering the use of drones for delivery to increase efficiency and respond to current customer needs. Consequently, consumer reactions to and perceptions of this new delivery method should also be analyzed. According to Farah et al. (2020) [3], despite efforts to position and consolidate drones as delivery service devices, consumers are skeptical about this innovation. Therefore, we should study how behavioral intentions towards drone food delivery services are formed after the COVID-19 outbreak [7,31].

### 2.1. Model and Hypotheses

Drones have shown great potential for parcel delivery both before [7] and after the pandemic [31,32]. However, the application of drones in food delivery services is not yet widely commercialized, as it is considered a novel technology in an emerging stage [32].

In the literature, the public acceptance of drones for goods delivery has been researched adopting the Diffusion of Innovation (DOI) theory and the Technology Acceptance Model (TAM) [31,49]. The DOI theory was proposed by Rogers (1983) [50] to understand why consumers adopt innovative technologies. He identified five attributes of innovations that could affect people's decision to adopt them: relative advantage, compatibility, complexity, observability, and trialability. Nevertheless, relative advantage, compatibility, and complexity have been the most commonly used [31].

Many times, the DOI theory has been applied in combination with the TAM proposed by Davis (1989) [51]. The TAM aims to explain how users come to accept the use of a certain technology based on a series of factors that influence their decision about how and when they will use it (e.g., perceived ease of use and usefulness as determinants of attitude (which in turn determines use) and external variables) [52]. Thanks to their similar constructions, the DOI and the TAM can complement each other. Furthermore, the attributes of innovation have often been considered to be determinants of attitude toward and intention to adopt certain technologies [53].

Yoo et al. (2018) [49] proposed a model that applies the DOI and TAM to formulate theoretical constructions and hypotheses. From Rogers' model (1983) [50], they took relative advantage, compatibility, complexity, and personal innovation as perceived technological factors. From Davis' model (1989) [51], they took the constructions attitude and intention to use. Finally, they evaluated perceived risks as an additional variable to those of the DOI and the TAM.

### 2.1.1. Relative Advantages

Relative advantage refers to the degree to which the consumer perceives that an innovation provides more benefits than the traditional tool or technology [54], in other words, its perceived superiority over the status quo and other options (e.g., for home package delivery). Nevertheless, said advantage can change due to different spatial characteristics or between cultures, beliefs, values, and other social dimensions. According to Rogers (1983) [50], this advantage is associated with a cost-benefit analysis to determine how convenient it is to adopt an innovation.

It has been found that the adoption of drones could be largely a matter of cost in relation to, e.g., helicopters [55] or traditional logistics systems [31]. Therefore, consumers perceive that drone delivery provides a relative advantage, which influences their attitude to adopt it thanks to its speed and environmental friendliness [31,49]. Park et al. (2018) [56] claim that the use of drones for food delivery is appropriate because they are fast and offer environmental benefits. Based on this information, this study proposes the first two hypotheses:

**Hypothesis 1.** *The relative advantage of speed positively affects attitude toward drone delivery.*

**Hypothesis 2.** *The relative advantage of environmental friendliness positively affects attitude toward drone delivery.*

### 2.1.2. Complexity

Complexity is defined as consumers' perception of technological advances and the ease of use of technologies [31,57]. According to Rogers (1983) [50], complexity is the extent to which an innovation is perceived by users as easy to use and understand. From a general point of view, innovations that are easier for consumers to use will be adopted more quickly; conversely, complex technologies may take longer or be rejected as they require new knowledge and development of skills [58]. The trialability of drones provides testing buffer prior to adoption, but their potential complexity is a concern that could hinder said adoption [22]. Therefore, a more generalized perception of less complexity in the use of drones for package delivery would influence their adoption [49]. As a result, the following hypothesis is proposed:

**Hypothesis 3.** *Lower complexity positively affects attitude toward drone delivery.*

2.1.3. Compatibility

Compatibility is a fundamental measure to predict or facilitate innovation, and it is defined depending on particular needs, values, and user experience [59]. Hence, it is assumed that people who find technologies to be compatible with their existing routines and needs are more likely to use them [60]. Regarding drone delivery, compatibility influences attitude, which plays an important role in the formation of behavioral intentions [61]. This is because when consumers evaluate new technologies, the overlap of perceived usefulness with perceived ease of use in the past positively affects their perceived compatibility and attitude toward new technologies [31]. Therefore, the following hypothesis is proposed:

**Hypothesis 4.** *Compatibility positively affects attitude toward drone delivery.*

2.1.4. Perceived Risks

Perceived risks have been used in a systematic way to try to explain and analyze the behavior of consumers in the face of new technologies; for instance, their anxiety in the face of unpleasant situations that they may experience when they buy new products or acquire new services, which are generally found in emerging fields [62].

Performance risk reflects consumers' concerns about losses incurred when a service does not work as expected; thus, they cannot make accurate performance decisions before using the service [45]. Consumers perceive a high performance risk in new products/services due to their lack of experience, which negatively affects their attitude toward them [62]. Thus, the following hypothesis is proposed:

**Hypothesis 5.** *Performance risk negatively affects attitude toward drone delivery.*

Delivery risk also reflects people's concerns about not getting a package delivered for a variety of reasons, such as an accident, damage, or theft of a drone carrying the package [45]. In addition, it is believed that drones might malfunction, perform inaccurate deliveries, or not find a place to land at residences [63]. Based on this, the following hypothesis is proposed:

**Hypothesis 6.** *Delivery risk negatively affects attitude toward drone delivery.*

Privacy risk refers to how much people value the confidentiality of their information, which directly influences their adoption of technologies. In the context of drone delivery, privacy is a driver of concern given the sensitivity of the information that may be collected [64]. This risk is related to the feeling of insecurity that individuals experience when they have to share personal data such as credit card number, address, and phone number [46]. Therefore, the following hypothesis is proposed:

**Hypothesis 7.** *Privacy risk negatively affects attitude toward drone delivery.*

2.1.5. Individual Characteristics

In general, individual characteristics are determinants of attitudes toward a technology [65], and individual innovativeness is a predominant factor in attitudes toward drone delivery [50,66]. This factor represents the degree to which a person feels open to using new technologies. Consequently, those with a great capacity for personal innovativeness are more likely to easily adopt new technologies and thus overcome the uncertainties that are generated in these processes. According to Ciftci et al. (2021) [66], this is a personality trait that drives an individual's initial intention to try innovations. Thus, the following hypothesis is proposed:

**Hypothesis 8.** *Personal innovativeness positively affects attitude toward drone delivery.*

Communication channels are especially useful to raise innovation awareness [67]. Mass media channels (e.g., the internet, television, radio, advertisements, and newspapers [49] inform individuals about new technologies [68]. During the pandemic, communication channels (especially social media) determined the acceptance and use of drones [69]. Thus, the following hypothesis is formulated:

**Hypothesis 9.** *Mass media channels positively affect attitude toward drone delivery.*

Over time, consumers' environmental awareness has increased, motivating the adoption and use of environmentally friendly—also called green—technologies. In the literature, environmental concerns have been related to the collective awareness of current environmental problems, according to Wu et al. (2019) [70], which can be indicated by the attitude, recognition, and response of individuals towards environmental problems. In particular, the adoption of drone delivery offers potential benefits for green consumers who believe that, by using this type of technology, are reducing their carbon footprint [64]. Therefore, the following hypothesis is proposed:

**Hypothesis 10.** *Environmental concern positively affects attitude toward drone delivery.*

### 2.1.6. Attitude and Intention

In the literature, it has been proposed and proven that attitude influences behavioral intentions, which is based on the ideas in the TAM. Consequently, behavioral intentions measure the probability of performing a certain action, such as adopting a technology [32]. Attitude refers to a person's positive or negative evaluation of a behavior, which has a direct effect on their intention to use [51]. In the case of drones, attitude is the negative or positive evaluation of their delivery service [31]. Based on this, the following hypothesis is presented:

**Hypothesis 11.** *Attitude toward drone delivery positively affects intention to use it.*

These eleven hypotheses (taken from [49]) compose the theoretical model adopted in this study to determine and analyze the factors that affect attitude toward drone delivery in Medellín, which in turn affects the intention to use said delivery in that city during the COVID-19 pandemic.

### 3. Materials and Methods

A survey was administered to 121 participants (15 and older) in Medellín. The participants were in different occupations and had knowledge of the existence of drones and some of their functions. The goal was to analyze the factors that affect their adoption of drone delivery in 2020 after the WHO declared a pandemic due to the outbreak of the COVID-19 virus. At that time, organizations were looking for service delivery strategies to face social distancing and lockdowns imposed to prevent the spread of the virus.

First, respondents were presented with the objective of this study. It was made clear to them that the survey was anonymous, they would not be paid or charged for participating in it, and they could be withdrawn from the study at any time. In the survey, drone delivery was connected to different significant sectors: food (home delivery), health (medicine delivery), and, in general, home delivery of online orders. The first part of its questionnaire included a total of 28 items designed to characterize the sample using open-ended questions about their interest in using drone delivery. The second part of the survey was a series of statements that participants rated on a Likert scale to measure the following constructs: Relative Advantage of Speed, Relative Advantage of Environmental Friendliness, Compatibility, and Complexity.

The first aim of this study was to apply and validate a model to examine the adoption of drone (unmanned aircraft) delivery in Medellín (a city in Colombia). The variables investigated here were selected from the model proposed by Yoo et al. (2018) [49], which includes the following eleven constructs: Attitude Towards Drone Delivery (ADD), Complexity (CX), Mass Media Channel (MMC), Compatibility (CM), Intention to Use Drone Delivery (IUD), Personal Innovativeness (PI), Delivery Risk (DR), Privacy Risk (PVR), Performance Risk (PMR), Relative Advantage of Environmental Friendliness (RAEF), and Relative Advantage of Speed (RAS). We designed the variables to extract the most useful information and thus achieve the aims of this study (see Table 1).

**Table 1.** Constructs and variables in the proposed model. The constructs were taken from [30].

| Construct | Variable |
| --- | --- |
| Attitude Towards Drone Delivery (ADD) | Drone delivery is easy to use. |
| | Using drones suits my lifestyle. |
| Complexity (CX) | My interaction with drone delivery is clear and understandable. |
| | Drone delivery can provide me with a better service. |
| | Using drone delivery fulfills my delivery service expectations. |
| Mass Media Channel (MMC) | I have a lot of information from the media about drone delivery. |
| | The media have helped me to better understand drone delivery. |
| Compatibility (CM) | Drones emit less carbon dioxide during delivery. |
| | Using drone delivery is compatible with all the aspects of my work. |
| Intention to Use Drone Delivery (IUD) | Using the drone delivery technology is a good idea. |
| | Receiving parcels delivered by drones is something that will happen in the long term. |
| Personal Innovativeness (PI) | I have often seen articles about drone parcel delivery. |
| | Drone delivery is desirable. |
| Delivery Risk (DR) | The package carried by the drone can be stolen. |
| | The package carried by the drone can be damaged by others. |
| Privacy Risk (PVR) | Drone delivery will result in a loss of my privacy. |
| | Drone delivery might be used in a way that violates my privacy. |
| Performance Risk (PMR) | The package carried by the drone might arrive late or be incomplete. |
| | Drone delivery will make me lose control over my privacy. |
| Relative Advantage of Environmental Friendliness (RAEF) | Drone delivery helps the environment. |
| | Drone delivery allows me to receive products in an environmentally friendly way. |
| Relative Advantage of Speed (RAS) | Drone delivery is a fast way to deliver packages. |
| | Drone technology is useful for fast goods delivery. |

Source: Yoo et al. [50].

In the survey, 45% of the participants were 30 or older, 31% were between 26 and 29, and the remaining percentage were between 16 and 25 years old. Additionally, 75% of those surveyed had never operated a drone in their lives, and the remaining 25% claimed that they had had an "excellent" or "very good" experiences with them. Among the participants, 55% would recommend buying a drone to their relatives and 68% thought that using drones is safe.

## 4. Results

IBM SPSS software was used to analyze and calculate the correlation statistics. Other values were also calculated: sampling adequacy measure, Bartlett's test of sphericity, standardized factor loadings, reliability of the measurement scale, and correlation between

the constructs in the model. The validity of the measurement scale was determined based on the analyses of convergent validity and discriminant validity. Such analyses had two aims: (1) to establish the reliability of the model based on the observable items and their impact on a latent variable and (2) to be able to claim that the measures of a single construct were valid; that is, that they were highly correlated to each other and could be discriminated from the measures proposed for a different construct [71].

### 4.1. Convergent Validity and Discriminant Validity

Principal component analysis was used for feature extraction. The factor loadings were obtained to interpret the function of every variable and define each one of the factors. The significant values reported in Table 2 determine that each variable adequately represents the factor that contains it. The guidelines to identify significant factor loadings were based on the sample size (121 participants), which accepts up to 0.50 in the value of each variable [72].

**Table 2.** Factor loadings of constructs in the proposed model.

| Factor | Item | Standardized Factor Loading | Average of Standardized Factor Loadings |
|---|---|---|---|
| Attitude Towards Drone Delivery (ADD) | ADD1 | 0.812 | 0.812 |
| | ADD2 | 0.812 | |
| Complexity (CX) | CX1 | 0.838 | 0.870 |
| | CX2 | 0.909 | |
| | CX3 | 0.863 | |
| Mass Media Channel (MMC) | MMC1 | 0.928 | 0.928 |
| | MMC2 | 0.928 | |
| Compatibility (CM) | CM1 | 0.761 | 0.761 |
| | CM2 | 0.761 | |
| Intention To Use Drone Delivery (IUD) | IUD1 | 0.900 | 0.900 |
| | IUD2 | 0.900 | |
| Personal Innovativeness (PI) | PI1 | 0.872 | 0.872 |
| | PI2 | 0.872 | |
| Delivery Risk (DR) | DR1 | 0.951 | 0.951 |
| | DR2 | 0.951 | |
| Privacy Risk (PVR) | PVR1 | 0.955 | 0.955 |
| | PVR2 | 0.955 | |
| Performance Risk (PMR) | PMR1 | 0.731 | 0.731 |
| | PMR2 | 0.731 | |
| Relative Advantage of Environmental Friendliness (RAEF) | RAEF1 | 0.886 | 0.886 |
| | RAEF2 | 0.886 | |
| Relative Advantage of Speed (RAS) | RAS1 | 0.967 | 0.967 |
| | RAS2 | 0.967 | |

Created using IBM® SPSS® Statistics.

Regarding the correlation between variables, Bartlett's test of sphericity and the Kaiser-Meyer-Olkin (KMO) measure of sampling adequacy were calculated, and the fit of the model was determined to carry out a factor analysis. The KMO is a statistical test that detects the correlation between variables and returns the probability that the correlation matrix contains significant values. Its *p*-value must be lower than the critical levels (0.05 or

0.01). Note that this test is very sensitive to increases in sample size because the larger the sample, the easier it is to find significant correlations [73].

Furthermore, the value of the KMO sampling adequacy measure (between 0 and 1) is defined as an index that compares the magnitudes of the observed correlation coefficients with those of the partial correlation coefficients. It characterizes those values on a scale in which KMO measures from 0.90 to 1.00 are marvelous; from 0.80 to 0.89, meritorious; from 0.70 to 0.79, middling; from 0.60 to 0.69, mediocre; from 0.50 to 0.59, miserable; and from 0.00 to 0.50, unacceptable [74]. Table 3 shows that the coefficients obtained by SPSS for each of the factors meet the criteria mentioned above, indicating that the data reduction technique can be applied.

**Table 3.** Sampling adequacy and Bartlett's test of sphericity of the factors in the proposed model.

| Factor | KMO Value | Bartlett Value | Meets Criteria |
|---|---|---|---|
| Attitude Towards Drone Delivery | 0.500 | 0.000 | Yes |
| Complexity | 0.697 | 0.000 | Yes |
| Maas Media Channel | 0.500 | 0.000 | Yes |
| Compatibility | 0.500 | 0.000 | Yes |
| Intention to Use Drone Delivery | 0.500 | 0.000 | Yes |
| Personal Innovativeness | 0.500 | 0.000 | Yes |
| Delivery Risk | 0.500 | 0.000 | Yes |
| Privacy Risk | 0.500 | 0.000 | Yes |
| Performance Risk | 0.500 | 0.000 | Yes |
| Relative Advantage of Environmental Friendliness | 0.500 | 0.000 | Yes |
| Relative Advantage of Speed | 0.500 | 0.000 | Yes |

Created using IBM® SPSS® Statistics.

The discriminant validity is evaluated in Table 4, which provides evidence of the confidence intervals of the model. Discriminant validity is one of the most common criteria used to evaluate scales for measuring latent constructs in social sciences. To prove the discriminant validity of the measures, those of the same construct must be highly correlated, and this correlation must be greater than that existing with respect to the measures proposed for any different construct [75].

**Table 4.** Confidence intervals of the variables in the model.

| | ADD | CX | MMC | CM | IUD | PI | DR | PVR | PMR | RAEF | RAS |
|---|---|---|---|---|---|---|---|---|---|---|---|
| ADD | . . . | | | | | | | | | | |
| CX | [0.179;0.576] | . . . | | | | | | | | | |
| MMC | [0.209;0.616] | [0.256;0.626] | . . . | | | | | | | | |
| CM | [0.367;0.684] | [0.208;0.579] | [0.075;0.493] | . . . | | | | | | | |
| IUD | [0.318;0.622] | [0.115;0.516] | [0.220;0.556] | [0.411;0.706] | . . . | | | | | | |
| PI | [0.312;0.687] | [0.261;0.631] | [0.489;0.754] | [0.325;0.656] | [0.274;0.595] | . . . | | | | | |
| DR | [0.406;0.030] | [0.263;0.141] | [0.292;0.088] | [0.094;0.505] | [0.077;0.328] | [0.063;0.397] | . . . | | | | |
| PVR | [0.152;0.567] | [0.065;0.338] | [0.143;0.483] | [0.111;0.489] | [0.021;0.415] | [0.139;0.263] | [0.094;0.312] | . . . | | | |

**Table 4.** *Cont.*

|  | ADD | CX | MMC | CM | IUD | PI | DR | PVR | PMR | RAEF | RAS |
|---|---|---|---|---|---|---|---|---|---|---|---|
| PMR | [0.712;0.345] | [0.200;0.218] | [0.194;0.557] | [0.115;0.554] | [0.164;0.559] | [0.002;0.440] | [0.473;0.751] | [0.473;0.751] | . . . | | |
| RAEF | [0.260;0.606] | [0.016;0.377] | [0.036;0.391] | [0.430;0.717] | [0.121;0.506] | [0.126;0.527] | [0.082;0.471] | [0.099;0.319] | [0.013;0.445] | . . . | |
| RAS | [0.127;0.555] | [0.184;0.603] | [0.126;0.306] | [0.187;0.618] | [0.127;0.507] | [0.208;0.603] | [0.206;0.227] | [0.027;0.379] | [0.042;0.406] | [0.227;0.641] | . . . |

Created using IBM® SPSS® Statistics.

In this study, the discriminant validity analysis was carried out by confirming that the confidence interval in the estimate of the correlation between each pair of factors did not contain a value of one [76].

*4.2. Realiability*

Next, we established the reliability of the measurement scale and verified the explanatory power of the model; for that purpose, we calculated the Cronbach's alpha of the respective scales of each construct. This procedure is necessary because Cronbach's alpha is an index used to measure the reliability of the internal consistency of a scale [77]. Its value ranges between 0 and 1, where numbers closer to 1 indicate a greater internal consistency of the items under analysis [78]. As shown in Table 5, the measurement instrument seems to have an adequate reliability of the internal consistency of the measurement scale because the value of the coefficients is within the range recommended by the authors mentioned above.

**Table 5.** Reliability coefficient.

| Factor | Cronbach's Alpha |
|---|---|
| Attitude Towards Drone Delivery | 0.811 |
| Complexity | 0.910 |
| Mass Media Channel | 0.943 |
| Compatibility | 0.745 |
| Intention to Use Drone Delivery | 0.912 |
| Personal Innovativeness | 0.890 |
| Delivery Risk | 0.963 |
| Privacy Risk | 0.964 |
| Performance Risk | 0.707 |
| Relative Advantage of Environmental Friendliness | 0.896 |
| Relative Advantage of Speed | 0.973 |

Created using IBM® SPSS® Statistics.

*4.3. Hypothesis Testing*

At the conceptual level, a factor analysis starts with previous hypotheses based on a given model. Then, the hypotheses are tested to determine the influence that certain variables have over others. The model proposed in this study was estimated to identify the determiners of the adoption of drone delivery in Medellín. The hypotheses formulated here were included in said model, and their degree of association was measured using Somers' D statistic.

Somers' D, which was used for this validation stage, is a measure that determines the strength and direction of the association between an ordinal dependent variable and an ordinal independent one. Thus, these ordinal variables contain a natural order that was measured on a Likert scale [79]. In this regard, the measure took values between −1 and 1,

where those close to 1 indicate a strong relationship between two variables (i.e., all pairs of the variables agree), and those close to −1 indicate that there is a week or no relationship between the constructs (i.e., all pairs of the variables disagree) [80].

Figure 1 presents the model proposed here and the Somers' D values obtained for the association between its constructs (i.e., variables). According to the theory reviewed in this study, we can conclude that the association coefficients calculated for the hypothetical relationships in the model present positive and significant values, which shows a high correlation between the variables evaluated in this analysis. In addition, SPSS provided the Somers' D coefficient and placed it in a cross tabulation to indicate the degree of association between the factors that were part of the hypotheses and those that were not. This enabled us not only to verify the degree of association of the hypothesized relationships but also to compare it with that between other constructs in the model.

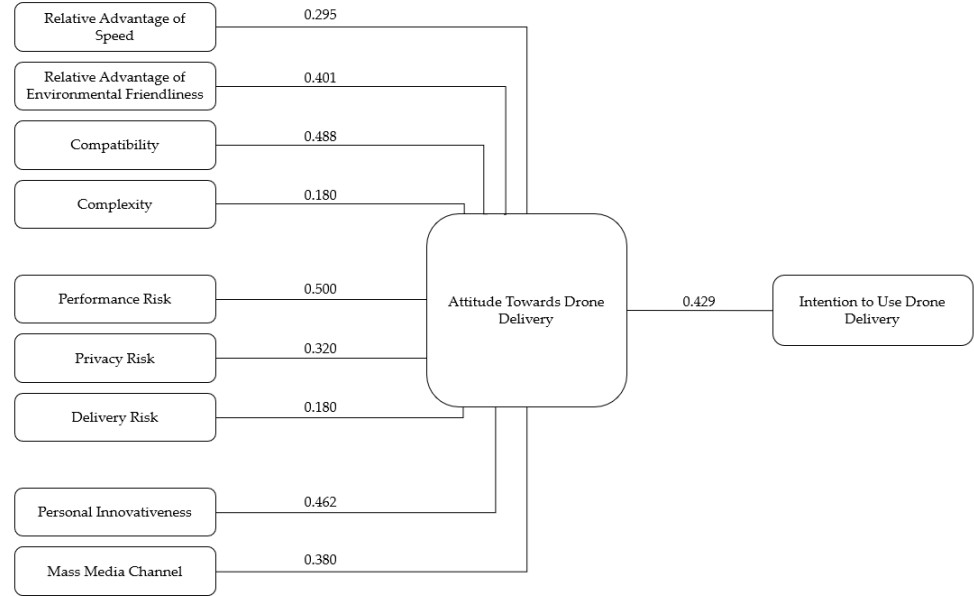

**Figure 1.** Model of adoption of drone delivery in Medellín.

## 5. Discussion

The results obtained for these hypothetical relationships show that Performance Risk has a significant correlation with Attitude Towards Drone Delivery, which is the strongest relationship in the model. This indicates that the possibility of an inconvenience in the provision of the service is a reason for consumers to perceive that the technical staff does not have complete control of the device when they send packages. This risk can generate great uncertainty in users and become a factor against the adoption of drones as a channel for goods delivery. These results coincide with those obtained by Yaprak et al. in 2021 [31] in the context of the COVID-19 pandemic. However, the construct Performance Risk can also be the key to change or improve perceptions of drone delivery because, if customers perceive its speed and efficiency, their satisfaction with the service is likely to increase and, thus, their attitude towards drone adoption is likely to improve.

The construct Personal Innovativeness has a strong association with Attitude Towards Drone Delivery, which is evidence that an individual's interest, level of curiosity, and conception of the delivery process have a positive impact on their Attitude Towards Drone Delivery. This construct is often one of the most influential in attitude towards drone delivery in emerging economies after the COVID-19 pandemic [44,81]. This is in line with Hwang et al. (2021) [82], who found that, under moderating effects, after the COVID-19 outbreak, consumers who were motivated to use drone food delivery services showed more favorable attitudes toward that new technology encouraged by social innovation. Consequently, organizations that provide drone delivery services should identify the

specific aspects that motivate consumer innovativeness in order to improve the efficiency of their services.

This claim is supported by the study of Yoo et al. (2018) [49], where the relative advantages of drone delivery, usability, perceived risks, and personal innovativeness were the main determinants of attitude towards drone delivery. Similarly, the results of the analysis by Kim et al. (2021) [7] highlighted the fundamental role of perceived innovativeness in the construction of consumer attitudes towards drone delivery services in the context of the COVID-19 pandemic.

Compatibility also exhibits a high level of association with Attitude Towards Drone Delivery. This result, which is supported by previous research [83], highlights the importance of compatibility for the adoption of flight technologies as means of packet transport. Hence, the organizations that support the implementation of delivery drones should further emphasize their compatibility on different media [84]. Therefore, Compatibility, an important factor according to the literature [85], influences Attitude Towards Drone Delivery.

Drones' Relative Advantage of Environmental Friendliness also presents a strong relationship with consumers' Attitude Towards Drone Delivery. The green image that delivery drones project (in terms of a small impact on the environment) favors a positive attitude towards their implementation. Likewise, previous studies have confirmed the advantages of environmentally friendly practices for shaping customers' attitude and, therefore, their intention to use drone delivery in emerging countries [44]. This may indicate that more and more people are transferring the principles of their lifestyle and their ethical and moral values to their decision-making process. If something deviates from their beliefs, they may not consider or approve it.

If individuals do not perceive that drone delivery contributes to the environment, their interest in using it may be reduced, and programs or actions that promote it may be undermined. Nevertheless, previous studies [8] have shown that drones will be able to optimize the way of eliminating contamination with a very high percentage (through the reduction of human contact) with the increase of the flexibility of the flight (reaching the less accessible regions every hour of the day).

Another significant relationship in the model was found between Attitude Towards Drone Delivery and Intention to Use Drone Delivery, which shows that the use of drone delivery technology should be associated with a positive feeling. These results are consistent with those of previous studies on consumer perception during the COVID-19 pandemic [31,82], which confirmed a positive relationship between attitude towards drone delivery service and intention to use that service. Individuals who are more inclined to be in favor of this technology consider that drone delivery is a good idea in the long term. However, prospective users should also feel that this technological reality is part of their lifestyle and not simply a utopian scenario they cannot be part of or benefit from in terms of product delivery.

Finally, the model presented a low correlation between the constructs Delivery Risk and Attitude Towards Drone Delivery. This indicates that, if their expectations are fulfilled and their needs are met, users tend to be more satisfied and motivated to continue using the services offered by drone delivery companies [31].

According to [31], there is a limited number of studies on order delivery using drones in times of the pandemic. Moreover, many studies on drone delivery have been conducted in developed economies, but only a few in their emerging counterparts [44], particularly in Latin America. The difference between this study and previous research is the context examined here, i.e., Medellín during lockdowns due to the COVID-19 pandemic declared by the WHO in 2020. This study revealed differences between findings obtained before and after the pandemic in developed economies and their emerging counterparts, but it focused on Medellín, Colombia, an emerging economy in Latin America. For that purpose, it tested the relationships between relative advantages, complexity, compatibility, perceived risks, individual characteristics, attitude, and intention in said city.

Before the pandemic, Yoo et al. (2018) [49] found that, in a developed country, relative advantage of speed, relative advantage of environmental friendliness, complexity, performance risk, privacy risk, and personal innovativeness were all significant predictors of attitude toward drones. In contrast, in this study, the most significant predictors were performance risk, compatibility, personal innovativeness, relative advantage of environmental friendliness, and control over the order. This indicates that, in an emerging economy (i.e., Medellín), the factors that influence drone delivery adoption are fast delivery, environmental friendliness, compatibility with lifestyles, performance of the technology device, home and personal data privacy, and orientation towards the use of innovative technologies. Contrary to the case analyzed by Yoo et al. (2018) [49], where customers were concerned about delivery speed, in Latin America they are more concerned about lack of control and product loss or damage caused by the drones during transport.

In the study by Yaprak et al. (2021) [31], the compatibility of drones was not influential in the context of the pandemic. This could be because the lifestyles of many people changed during the lockdowns, and new ways to meet people's daily needs were adopted. Other authors have paid attention to consumers in developed economies and their perception of the benefits and risks of drones. In emerging economies, personal innovativeness tends to be ranked higher than consumer attitudes and environmental friendliness with respect to drone adoption [44]. Considerable attention has also been paid to opinion passing and perceived privacy risk [83], and the results largely coincide with those obtained in this study.

Regarding theoretical implications, the results of this study provide empirical evidence of the robustness of the model proposed by Yoo et al. (2018) [49]. They also indicate that Performance Risk, Compatibility, Personal Innovativeness, and Relative Advantage of Environmental Friendliness are the most influential factors on Intention to Use Drone Delivery (mediated by Attitude Towards Drone Delivery). In addition, this study paves the way for future research in this area in Latin America after the pandemic because drones have become an innovative technology for parcel delivery and have proven to be very useful in the context of a pandemic. Indeed, not many studies have been published in this field. Therefore, this study contributes to the emerging line of research on the adoption of drone delivery in emerging economies in Latin America. Furthermore, it highlights the relevant role of performance risk, compatibility, innovativeness, and relative advantage of environmental friendliness in a positive attitude toward the use of drones. Participants' decisions are greatly influenced by concerns about theft or damage, lifestyles, early adoption of innovations, and the trend of environmentally friendly technologies. Although these results are not unexpected, they provide additional information about a lockdown scenario that was not considered in the original TAM. Future studies could address other important factors in the literature (e.g., ease of use and perceived usefulness) in other cities in Colombia or other Latin American countries.

In terms of practical implications, this paper can provide an input for decision-making by companies interested in adopting this technology for commercial purposes. Thus, they can consider the factors that affect user attitudes to refine their drone delivery systems. Based on the results obtained, organizations can find a way to promote the use of this type of service so that customers have reason to believe that drone package delivery is innovative, safe, and environmentally friendly. They can also identify the perceived risks that generate the greatest concern in consumers to act and disseminate relevant information on the matter. Thus, to enjoy these benefits and scale business drone operations after the pandemic, drone delivery services should be geared towards improving convenience with proper packaging, tracking, and trouble-free deliveries, as well as faster delivery times, lower costs (to attract a larger number of consumers), and environmental advantages. In general terms, this study is valuable for decision-makers at organizations that provide online shopping services and are working on the implementation of drone delivery as a means of transporting packages.

The limitations of this study are present in three aspects. First, it is necessary to consider the current lack of knowledge of all the different possible uses of drones, especially for goods distribution, which could greatly affect the adoption of this type of technology. Second, trust is important for drone delivery, especially in developing countries. Consequently, distrust due to security and privacy issues may delay drone adoption in the delivery market in said countries. Third, the sample size was not large enough to generalize the findings to the overall adoption of drones to distribute goods in Medellin. A larger sample is needed to obtain more generalizable results.

Investigating the adoption of a technology implementing technology acceptance models reduces technical, operational, and organizational uncertainty for developer companies. Technology adoption was accelerated during the COVID-19 pandemic and much more so in the post-pandemic context. This pressing need is forcing different sectors to acknowledge emerging technological capabilities. For instance, drones have potential to transform industries and improve productivity. Importantly, the data collected in this study highlight key elements to foster innovation.

Innovative initiatives combine research, collaborative work, needs, resources, and the market, among other aspects. Thus, identifying the particular factors that influence the adoption of drones for goods delivery reduces uncertainty for organizations because they can use specific constructs to guide product development or corporate process innovation.

This study presented theoretical information about drone delivery adoption, but knowledge generation in this field is still limited. Participants in the survey were concerned about technical aspects of drones; however, they were open and willing to use drone delivery if it improves their quality of life. Finally, they were also concerned about drones' performance risk, which means that knowledge dissemination campaigns should be implemented to highlight the advantages and possible integration of drone delivery services.

## 6. Conclusions

The COVID-19 pandemic has generated the need to reduce the risk of infection using various self-care strategies such as social distancing. Even after the pandemic, some changes that were implemented as preventive measures will remain in force. Such is the case of drone delivery, which was already being developed before the outbreak and had attracted the interest of scholars and companies that provide this commercial service.

Multiple organizations have made efforts to implement contactless delivery strategies, but users' attitude is vital in the implementation of these technologies as a means of delivery. Therefore, the aim of this study was to apply and validate a model to identify the determinants of the adoption of drones (unmanned aircraft) to deliver goods in Medellín in the context of the COVID-19 pandemic. The knowledge of the consumer attitudes that influence the acceptance of these technologies has positive applications in academic and commercial contexts.

This study proposed and applied a model in which relative advantage is a multidimensional construct. It also investigated the determinants that directly influence consumers' attitude towards and intention to adopt drone delivery services by using three types of variables: (1) perceived attributes, (2) perceived risks, and (3) individual characteristics. Compatibility and Relative Advantage of Environmental Friendliness (perceived attributes); Performance Risk (a perceived risk); and Personal Innovativeness (an individual characteristic) exhibited the strongest influence on Attitude Towards Drone Delivery in this model applied in the context of the COVID-19 pandemic in Medellín.

**Author Contributions:** Conceptualization, A.V.-A. and P.A.R.-C.; methodology, P.A.R.-C., A.V.-A. and G.M.-L.; software, P.A.R.-C. and J.D.L.C.-V.; validation, M.B.-A. and G.M.-L.; formal analysis, M.B.-A.; investigation, J.C.P.-V.; resources, P.A.R.-C.; data curation, J.C.P.-V.; writing—original draft preparation J.C.P.-V., M.B.-A. and J.D.L.C.-V.; writing—review and editing, A.V.-A.; visualization, J.D.L.C.-V.; supervision, M.B.-A. and G.M.-L.; project administration, A.V.-A.; funding acquisition, P.A.R.-C. and J.D.L.C.-V. All authors have read and agreed to the published version of the manuscript.

**Funding:** This project received external funding from the following universities Instituto Tecnológico Metropolitano, Institución Universitaria Escolme, and Institución Universitaria Marco Fidel Suárez (Medellín, Colombia). We would like to thank ITM Translation Agency for language editing the original manuscript.

**Institutional Review Board Statement:** The study was conducted according to the guidelines of the Declaration of Helsinki, and ap-proved by the Ethics Committee of Institución Universitaria Escolme (protocol code PC2022-03.06.2021).

**Informed Consent Statement:** Informed consent was obtained from all subjects involved in the study.

**Data Availability Statement:** Not applicable.

**Conflicts of Interest:** The authors declare no conflict of interest.

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
