# Peer review of "Factors Associated with the Adoption of Drones for Product Delivery in the Context of the COVID-19 Pandemic in Medellín, Colombia"

_drones, doi:10.3390/drones6090225_

Round 1

Reviewer 1 Report

The proposed paper is addressing an important area regarding public attitudes towards the increasing development of drone delivery services in everyday life. While there has been extensive statistical analysis of the data obtained there are some areas of the paper that could be tighter as outlined below.

Introduction:

Line 46 - There are many instances of health service delivery - Zipline in Africa is just one example.

There is no background about the case study city Medellin and why it is a suitable choice for the survey. Further how has Covid-19 affected the city and how might drone delivery services improve life there? 

Are drones used in other capacities in the city (agriculture, leisure, commercial etc) and therefore how familiar are local people with drones? The context in which drones are used for delivery is crucial - trying to implement drones as a friendly delivery service in a region where they have been used to cary or drop bombs on civilians will change peoples perceptions. The theory on this has been explored by Gregoire Chamayou, 2013 (A theory of the drone) and Peckham and Sinha, 2019 (Anarchitectures of Health: Futures of the Biomedical Drone).

In the Materials and Methods section more information should be given about the selection criteria of participants and conduction of the survey. Are they regular citizens, business owners, age, gender...

One thing that I would like to see made clearer is what type of potential drone delivery services the survey participants were responding to. Health delivery services (essential services) might have a different answer profile to pizza delivery (convenience) for example.

Covid-19 relevance - to validate utilising Covid-19 as a pretext for the survey there needs to be pre Covid-19 comparative data as a baseline.

Why was qualitative data from the "open-ended questions" not presented? I think this is just as important if not more so than statistical analysis. Qualitative statements might have been better used to back up the statistics, particularly on issues of concern such as privacy and delivery risks.

The conclusion comes back again to Covid-19 but I don't think this has been tested specifically, this seems more of a general assumption. Overall the authors need to decide whether the data fits the title of the paper.

In summary the paper would benefit from some reorganisation and depth.

Author Response

Medellín, August 18, 2022

Dear Diego González-Aguilera

Editor-in-Chief

Drones

Kind regards

According to the review of our article by the Reviewer 1, the following changes were made, properly marked with red letters in the article:

Reviewer

Comment

Response

Reviewer #1

Introduction - Line 46 - There are many instances of health service delivery - Zipline in Africa is just one example.

Information on cases of Zipline health service provision in Africa is included in lines 50-52.

Reviewer #1

Introduction: There is no background about the case study city Medellin and why it is a suitable choice for the survey.

Background on the use of drones in different services in the city case study is included in line 53-68. Therefore, a survey is suitable for the collection and measurement of data to test the relationship between the variables of the model used.

Reviewer #1

Introduction: How has Covid-19 affected the city and how might drone delivery services improve life there? 

Information is included on how the COVID-19 pandemic has affected the city's population and how drone services could help with this situation in lines 105-120.

Reviewer #1

Introduction: Are drones used in other capacities in the city (agriculture, leisure, commercial etc) and therefore how familiar are local people with drones? The theory on this has been explored by Gregoire Chamayou, 2013 (A theory of the drone) and Peckham and Sinha, 2019 (Anarchitectures of Health: Futures of the Biomedical Drone).

At the moment, the use of the drone service for the monitoring and mitigation of landslide risks in low-income settlements in the city is registered in the city in academic research, as well as support in the collection of data for thermal analysis of urban environments in the Valley of Bored. This information is included in the introduction in lines 53-60.

Reviewer #1

Material and methods: Add more information should be given about the selection criteria of participants and conduction of the survey. Are they regular citizens, business owners, age, gender...

More information about the people surveyed is included. People over 16 years old with any occupation, inhabitant of the city of Medellín with knowledge about the existence of drones and some of their functionalities were selected in lines 335-340.

Reviewer #1

Materials and methods: What type of potential drone delivery services the survey participants were responding to. Health delivery services (essential services) might have a different answer profile to pizza delivery (convenience) for example.

The requested information is included in lines 341-345.

Reviewer #1

Covid-19 relevance - to validate utilising Covid-19 as a pretext for the survey there needs to be pre Covid-19 comparative data as a baseline.

Reference is made to comparative studies in the same context before and after the pandemic in introduction section.

Reviewer #1

Why was qualitative data from the "open-ended questions" not presented?

Are included in lines 363-368.

Reviewer #1

Conclusion: The conclusion comes back again to Covid-19, but I don't think this has been tested specifically, this seems more of a general assumption.

Information based on a general assumption is excluded.

Reviewer #1

Title: Overall the authors need to decide whether the data fits the title of the paper.

It is recommended to leave the same title because it is related to the object and the data found.

Reviewer #1

Abstract: In summary the paper would benefit from some reorganisation and depth.

Information about the models and the findings found are included in lines 15-17.

We look forward to your comments and hope to hear from you soon.

Thank you very much

_

The authors

Reviewer 2 Report

Factors associated with the adoption of drones for product delivery in the context of the COVID-19 pandemic in Medellín, Colombia

This study is interesting in that it focuses on drones that are currently the biggest issue in the diverse industries. There are limitations of the adoption of drones for product delivery, but I think the implications are not bad. The opinions and comments are illustrated as follows.

First, the reason for introducing drones for product delivery is unclear. For example, the authors mentioned that the introduction of drones for product delivery has accelerated due to COVID-19, but argue that drones should be introduced to reduce costs and increase work efficiency.

Second, in the current technological situation, drones are difficult to completely replace the current systems of delivery services. The authors must acknowledge this part.

Third, the introduction is too lengthy. It needs to be simplified. 

Fourth, in the introduction part, it is necessary to emphasize the difference from existing studies.

Fifth, it is also necessary to explained perceived risks of drone food delivery services (e.g. financial risk, time risk, privacy risk, performance risk, and psychological risk).

Sixth, data analysis was performed properly and well.

Seventh, implications for the data analysis results are very insufficient. In particular, the authors need to think a lot about practical implications.

Author Response

Medellín, August 18, 2022

Dear Diego González-Aguilera

Editor-in-Chief

Drones

Kind regards

According to the review of our article by the Reviewer 2, the following changes were made, properly marked with red letters in the article:

Reviewer

Comment

Response

Reviewer #2

Introduction: The reason for introducing drones for product delivery is unclear.

To justify the reason for the introduction of drones for the delivery of goods, it is explained that this serves as a contagion prevention measure in the delivery of goods at home in the city of Medellín. Also, it is left explicit in the research objective in lines 53-60.  

Reviewer #2

Introduction: In the current technological situation, drones are difficult to completely replace the current systems of delivery services. The authors must acknowledge this part.

This part is recognized and some barriers to implementation of drones in logistics are warned in lines 149-159.

Reviewer #2

Introduction: The introduction is too lengthy. It needs to be simplified. 

The introduction is shortened, and a literature review section is included.

Reviewer #2

Introduction: It is necessary to emphasize the difference from existing studies.

With respect to existing studies, the difference is highlighted, since the registered studies have been applied in developed countries, while this study is applied in an emerging economy. Lines 105-110

Reviewer #2

Introduction: It is necessary to explained perceived risks of drone food delivery services (e.g. financial risk, time risk, privacy risk, performance risk, and psychological risk).

The perceived risks of drone food delivery services are explained: performance, delivery, privacy, financial and psychological in lines 168 – 176 and 272-275.

Reviewer #2

Implications for the data analysis results are very insufficient. In particular, the authors need to think a lot about practical implications

The information on the implications of the article is expanded in lines 577-587.

We look forward to your comments and hope to hear from you soon.

Thank you very much

_

The authors

Reviewer 3 Report

Dear Authors,

The paper addresses an important and currently discussed topic of drone usage in parcels delivery which corresponds with the scope of the Drones journal. However, its quality is very poor and scientific soundness is very low. The paper lacks a crucial part - literature review, the research process is incomplete, the method and results are presented in dissatisfying way, the discussion part is limited as it is  based only on a few previous papers. Therefore, in my opinion, the paper cannot be published in present form. Below I enclose more remarks:

1.     Regarding the topic of article, the introduction (lines 48-53 ) should include the current data about e-commerce market, especially its growth during pandemic, while the authors cite outdated data from 3 years ago. It would be interesting to present the examples of drone usage in different parts of the world.

2.     The introduction should briefly place the study in a broad context and present the research gap, highlight why this work is important and define the significance of work.

3.     The article lacks of the well-developed literature review of similar research. The broad review of models used to analyse customer acceptance of technologies should be presented, while the authors present only one of them.

4.     In Materials and Method Section the authors present constructs and variables in the proposed model. In this section the theoretical model should be presented in order to be confirmed in this research. On the basis of literature review, the authors should formulate the hypothesis, which should verified in this research.

5.     The authors do not explain what is their contribution into the theory, if they used the original model of different authors (Yoo et al. , 2018). There is no information what are new aspects/elements of proposed model.

6.     The characteristics of respondents should be presented in Materials and Method Section. There is no information about the way of respondence selection and if the research sample was representative.

7.     The Discussion part is very poorly developed, as it contains the analysis and comparison of only four previous papers. The findings and their implications should be discussed in the broadest context possible. Future research directions may also be highlighted.

8.   There is a lot of typos, technical mistakes e.g. source of data in tables should be indicated below the table; the Author Contribution section should be filed in properly.

9.     The Literature section is very limited, as it contains only 25 papers.

Author Response

Medellín, August 18, 2022

Dear Diego González-Aguilera

Editor-in-Chief

Drones

Kind regards

According to the review of our article by the Reviewer 3, the following changes were made, properly marked with red letters in the article:

Reviewer

Comment

Response

Reviewer #3

Introduction: Lines 48-53, should include the current data about e-commerce market, especially its growth during pandemic, while the authors cite outdated data from 3 years ago.

A reference to the year 2022 is added that shows, from an RSL, the growth of articles on e-commerce logistics in lines 71-74.

Reviewer #3

Introduction:  It would be interesting to present the examples of drone usage in different parts of the world.

The use of drones to deliver medicines and samples to suspected COVID-19 patients in Africa is exposed. It is exposed how drones have been used in the city of Medellín to analyze the risks of landslides in low-income settlements and to collect data from urban heat islands. In the context of the pandemic, it is exposed how the use of these have helped to reduce the risk of infection of doctors or other medical personnel, also in disinfection in outdoor public service areas, as well as for monitoring and detection, social distancing, disinfection, data analysis, delivery of goods and medical supplies. As well as for home delivery of online orders, to reduce contact between people as a protective measure. Lines 50-68 and 105-126.

Reviewer #3

Introduction: Should briefly place the study in a broad context and present the research gap, highlight why this work is important and define the significance of work.

From the examples provided in the literature, the importance of the study is highlighted, and the research objective is established. The research gap is also established. Lines 105-126.

Reviewer #3

Material and methods: The article lacks of the well-developed literature review of similar research

References with a Background of similar studies and the use of drones for other services before and during the COVID-19 pandemic are included.

Reviewer #3

Results: The broad review of models used to analyse customer acceptance of technologies should be presented, while the authors present only one of them.

Background on the models used to analyze the adoption of these technologies is presented in lines 206- 333.

Reviewer #3

Materials and Method: Theoretical model should be presented in order to be confirmed in this research.

The theoretical model is presented, as well as the research hypotheses in the subsection: Model and hypotheses. Lines 206- 333.

Reviewer #3

Materials and Method: On the basis of literature review, the authors should formulate the hypothesis, which should verified in this research.

A literature review is made to justify the proposed research hypotheses. Lines 206- 333.

Reviewer #3

Discussion: The authors do not explain what is their contribution into the theory, if they used the original model of different authors (Yoo et al., 2018). There is no information what are new aspects/elements of proposed model.

With respect to the original model, this study contributes from the perspective of an emerging economy in Latin America (where studies of this type have not yet been recorded) and is compared with the results obtained in developed economies before and after the pandemic. This is included in the document. Lines 475-482; 490-493; 496-501; 513-516; 525-555; 564-573; 577-614.

Reviewer #3

Materials and method: The characteristics of respondents should be presented in Materials and Method Section. There is no information about the way of respondence selection and if the research sample was representative.

Information on the selection of study participants, as well as their characteristics, is included. Sampling is non-probabilistic based on criteria (already exposed). Lines 335-345.

Reviewer #3

Discussion: is very poorly developed, as it contains the analysis and comparison of only four previous papers

More theoretical references are added to contrast the results. Lines 475-482; 490-493; 496-501; 513-516; 525-555; 564-573; 577-614.

Reviewer #3

Results: The findings and their implications should be discussed in the broadest context possible.

Implications of the study are expanded. Lines 564-587.

Reviewer #3

Future research directions may also be highlighted.

It is included.

Reviewer #3

Results: Source of data in tables should be indicated below the table

Data source is added in the 5 tables contained in the document

Reviewer #3

Discussion: The Author Contribution section should be filed in properly.

Information is included on the contribution of the research from the research approach

Reviewer #3

Introduction: The Literature section is very limited, as it contains only 25 papers.

New additional references are added in the introductory part, highlighted in red for easy identification.

We look forward to your comments and hope to hear from you soon.

Thank you very much

_

The authors

Round 2

Reviewer 2 Report

No more comment 

Reviewer 3 Report

Although almost all my remarks were corrected, but regarding the quality of journal, the novelty and contribution of this study is not sufficient to publish the result. Another main problem is too small research sample, so the obtained results do not reflect the acceptence level of drones by society in Medellin.